# Constrained Latent Action Policies for Model-Based Offline Reinforcement Learning

**Marvin Alles**[1,2]*  **Philip Becker-Ehmck**[1]  **Patrick van der Smagt**[1,3]  **Maximilian Karl**[1]

[1]Machine Learning Research Lab, Volkswagen Group   [2]Technical University of Munich
[3]Eötvös Loránd University Budapest
{marvin.alles, philip.becker-ehmck, maximilian.karl}@volkswagen.de

## Abstract

In offline reinforcement learning, a policy is learned using a static dataset in the absence of costly feedback from the environment. In contrast to the online setting, only using static datasets poses additional challenges, such as policies generating out-of-distribution samples. Model-based offline reinforcement learning methods try to overcome these by learning a model of the underlying dynamics of the environment and using it to guide policy search. It is beneficial but, with limited datasets, errors in the model and the issue of value overestimation among out-of-distribution states can worsen performance. Current model-based methods apply some notion of conservatism to the Bellman update, often implemented using uncertainty estimation derived from model ensembles. In this paper, we propose Constrained Latent Action Policies (C-LAP) which learns a generative model of the joint distribution of observations and actions. We cast policy learning as a constrained objective to always stay within the support of the latent action distribution, and use the generative capabilities of the model to impose an implicit constraint on the generated actions. Thereby eliminating the need to use additional uncertainty penalties on the Bellman update and significantly decreasing the number of gradient steps required to learn a policy. We empirically evaluate C-LAP on the D4RL and V-D4RL benchmark, and show that C-LAP is competitive to state-of-the-art methods, especially outperforming on datasets with visual observations.[1]

## 1   Introduction

Deep-learning methods are widely used in applications around computer vision and natural language processing, related to the fact that datasets are abundant. But when used for control of physical systems, in particular with reinforcement learning, obtaining data involves interaction with an environment. Learning through trial-and-error and extensive exploration of an environment can be done in simulation, but hard to achieve in real world scenarios [1, 2, 3]. Offline reinforcement learning tries to solve this by using pre-collected datasets eliminating costly and unsafe training in real-world environments [4, 5, 6].

Using online reinforcement learning methods in an offline setting often fails. A key issue is the *distributional shift*: the state-action distribution of the offline dataset, driven by the behavior policy, differs from the distribution generated by a learned policy. This leads to actions being inferred for states outside the training distribution. Therefore, value-based methods are prone to overestimating values due to evaluating policies on out-of-distribution states. This leads to poor performance and unstable training because of bootstrapping [6, 7, 8]. Offline reinforcement learning methods address

---

*Corresponding author.
[1]Code is available at: `https://github.com/marvinalles/c-lap`

38th Conference on Neural Information Processing Systems (NeurIPS 2024).

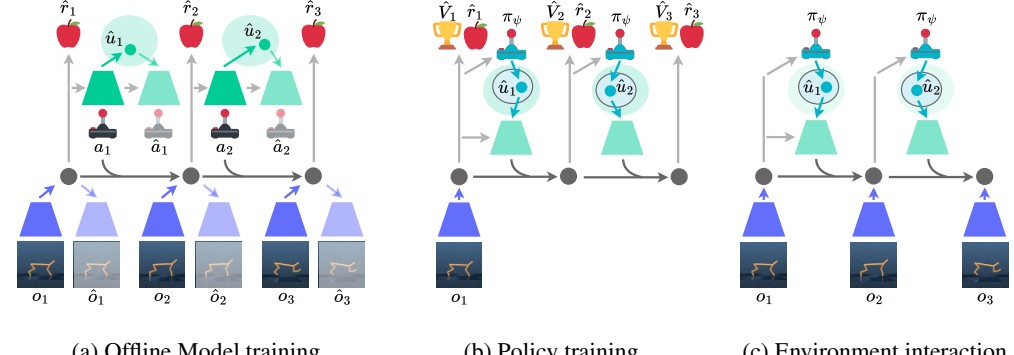

(a) Offline Model training      (b) Policy training      (c) Environment interaction

Figure 1: Overview of C-LAP. (a) The model is trained offline. It is encoding observations $o_t$ and actions $a_t$ to latent states (gray circle) and latent actions $u_t$ (green circle), and decoding them thereafter. Furthermore it is predicting rewards $\hat{r}_t$. (b) The policy is learned in the latent action space, but constrained to the support of the latent action prior, and uses the generative capabilities of the action decoder. Gradients are computed by back-propagating estimated values $\hat{V}_t$ and rewards $\hat{r}_t$ through the imagined trajectories. (c) The policy is used in the real world, again using the generative action decoder.[1]

this issue with different approaches and can be categorized into model-free and model-based methods, similar to online reinforcement learning.

Model-*free* offline reinforcement learning usually follows one of the following paradigms: constrain the learned policy to the behavior policy [9, 10, 11, 7]; or introduce some kind of conservatism to the Bellman update [12, 13, 14, 8]. Model-*based* reinforcement learning methods transform the offline to an online learning setting: They approximate the system dynamics and try to resolve the evaluation of out-of-distribution states by using the generalization capabilities of the model and generating additional samples. But as the training distribution is fixed, the estimation capabilities of the model are limited. Therefore, these model-based methods also rely on a conservative modification to the Bellman update as a measure to counteract value overestimation which is mostly achieved through uncertainty penalties [15, 16, 17, 18, 19, 20, 21]. Apart from the typical approach of using an auto-regressive model to estimate the dynamics, other model-based methods treat the objective as trajectory modeling [22, 23, 24]. These methods aim to combine decision making and dynamics modeling into one objective. Instead of learning a policy, they sample from the learned trajectory model for planning. We will refer to the first kind of methods, which learn a dynamics model to train a policy, as model-based reinforcement learning.

We aim to solve the problem of value overestimation in model-based reinforcement learning by jointly modeling action and state distributions, without the need for uncertainty penalties or changes to the Bellman update. Instead of learning a conditional dynamics model $p(s \mid a)$, we estimate the joint state-action distribution $p(s, a)$. This is similar to methods that frame offline reinforcement learning as trajectory modeling, but we use an auto-regressive model and still learn a policy. By formulating the objective as a generative model of the joint distribution of states and actions, we create an implicit constraint on the generated actions, similar to [10, 25]. The goal of this approach is to address the shift in the entire distribution, rather than looking at out-of-distribution actions and states separately. We achieve this using a recurrent state-space model with a latent action space, which we call the recurrent latent action state-space model. Using a latent action space allows us to learn a policy that uses the latent action prior as an inductive bias. This approach keeps the policy close to the original data while allowing it to change when needed, which makes learning the policy much faster. To achieve this, we treat policy optimization as a constrained optimization problem, similar to enforcing a support constraint [9]. We provide a high level overview of our method in Figure 1.

---

Overall, we summarize our contribution as follows:

- We introduce latent action state-space models for model-based offline reinforcement learning, treating it as auto-regressive generative modeling of the joint distribution of states and actions.
- We formulate policy optimization as a constrained optimization problem, using the latent action space to generate actions within the support of the dataset's action distribution and jump-start policy learning by using the generative action decoder.
- We evaluate our approach on one benchmark with image observations (V-D4RL [20]) and on another one with low-dimensional feature observations (D4RL [26]).
- We evaluate the effect of our approach on value overestimation.

## 2 Preliminaries

We consider a partial observable Markov decision process (POMDP) defined by $\mathcal{M} = (\mathcal{S}, \mathcal{A}, \mathcal{O}, T, R, \Omega, \gamma)$ with $\mathcal{S}$ as state space, $\mathcal{A}$ as action space, $\mathcal{O}$ as observation space, $s \in \mathcal{S}$ as state, $a \in \mathcal{A}$ as action, $o \in \mathcal{O}$ as observation, $T : \mathcal{S} \times \mathcal{A} \to \mathcal{S}$ as transition function, $R : \mathcal{S} \to \mathbb{R}$ as reward function, $\Omega : \mathcal{S} \to \mathcal{O}$ as emission function and $\gamma \in (0, 1]$ as discount factor. The goal is to find a policy $\pi : \mathcal{O} \to \mathcal{A}$ that maximizes the expected discounted sum of rewards $\mathbb{E}[\sum_{t=1}^{T} \gamma^t r_t]$ [27].

In online reinforcement learning, an agent iteratively interacts with the environment $\mathcal{M}$ and optimizes its policy $\pi$. In offline reinforcement learning, however, the agent cannot interact with the environment and must refine the policy using a fixed dataset $\mathcal{D} = \{(o_{1:T}, a_{1:T}, r_{1:T})_{n=1}^{N}\}$. Therefore, the agent must understand the environment using limited data to ensure the policy maximizes the expected discounted sum of rewards when deployed [6]. Auto-regressive model-based offline reinforcement learning tries to learn a parametric function to estimate the transition dynamics $T$. The transition dynamics model is then used to generate additional trajectories which can be used to train a policy. The majority of these approaches learn a dynamics model directly in observation space $T_\theta(o_t \mid o_{t-1}, a_{t-1})$ [16, 17, 18, 19, 28], while others use a latent dynamics model $T_\theta(s_t \mid s_{t-1}, a_{t-1})$ [20, 21].

## 3 Constrained Latent Action Policies

A main issue in offline reinforcement learning is value overestimation, which we address by ensuring the actions generated by the policy stay within the dataset's action distribution. Unlike previous model-based methods, we formulate the learning objective as a generative model of the joint distribution of states and actions. We do this by combining a latent action space with a latent dynamics model. Next, we use the generative properties of the action space to constrain the policy to the dataset's action distribution. A general outline of our method, Constrained Latent Action Policies (C-LAP ), is shown in Appendix B. It starts with learning a generative model, followed by actor-critic agent training on imagined trajectories, similar to the methods in [29, 30, 20, 21].

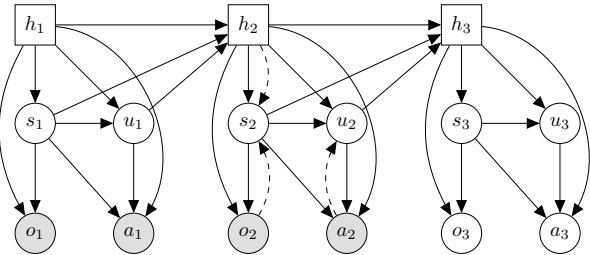

Figure 2: Recurrent latent action state-space model. The generative process is shown by solid lines and inference by dashed lines. Stochastic variables are denoted by circles and deterministic variables by rectangles.

**Generative model** Model-based offline reinforcement learning requires learning a model that is accurate in areas with low data coverage but also generalizes well. Therefore, it's crucial to balance

staying within the dataset's distribution and generalizing to unseen states. We propose a generative model

$$p(o_{1:T}, a_{1:T}) = \int p(o_{1:T}, a_{1:T} \mid s_{1:T}, u_{1:T}) p(s_{1:T}, u_{1:T}) \, ds \, du. \tag{1}$$

that jointly models the observation and action distribution of a static dataset $\mathcal{D} = \{(o_{1:T}, a_{1:T}, r_{1:T})_{n=1}^{N}\}$ by using latent states $s_t$ along with latent actions $u_t$. Unlike other model-based offline reinforcement learning methods that learn a conditional model $p(o_{1:T} \mid a_{1:T})$ and rely on ensemble based uncertainty penalties on the Bellman update to generate trajectories within the data distribution [21, 18, 16, 17], our approach uses a latent action space to impose an additional implicit constraint. By implementing a policy in the latent action space, generated actions will stay within the dataset's action distribution, thus enabling generalization within the limits of the learned model [10]. We empirically validate this claim in Appendix F. To obtain a state space model with Markovian assumptions on the latent states $s_t$ we impose the following structure:

$$p(o_{1:T}, a_{1:T} \mid s_{1:T}, u_{1:T}) = \prod_{t=1}^{T} p(o_t \mid s_t) p(a_t \mid s_t, u_t), \tag{2}$$

$$p(s_{1:T}, u_{1:T}) = \prod_{t=1}^{T} p(u_t \mid s_t) p(s_t \mid s_{t-1}, u_{t-1}). \tag{3}$$

We implement the probabilistic model modifying the design of a recurrent state-space model [31]. Thus, the latent dynamics model $p(s_t \mid s_{t-1}, u_{t-1})$ is based on the deterministic transition $f(h_{t-1}, s_{t-1}, a_{t-1})$ using the latent action decoder $p_\theta(a_{t-1} \mid s_{t-1}, u_{t-1})$ to generate actions. In the following, we mostly omit deterministic states $h_t$ for notational brevity. The resulting recurrent latent action state-space model is shown in Figure 2 and consists of the following components, specifically

| | |
|---|---|
| latent state prior | $p_\theta(s_t \mid s_{t-1}, u_{t-1})$, |
| latent action prior | $p_\theta(u_t \mid s_t)$, |
| observation decoder | $p_\theta(o_t \mid s_t)$, |
| and action decoder | $p_\theta(a_t \mid s_t, u_t)$. |

The latent state prior predicts the next latent state $s_t$ given the previous latent state $s_{t-1}$ and action $u_{t-1}$ using the deterministic transition and the action decoder. The latent action prior predicts latent actions $u_t$ given latent state $s_t$. Latent states as well and as latent actions are decoded using their respective decoder. Similar to [25, 10] actions are reconstructed given latent state and latent action.

Directly maximizing the marginal likelihood is intractable, hence we maximize the evidence lower bound (ELBO) on the log-likelihood $\log p(o_{1:T}, a_{1:T})$ instead. To approximate the true posterior, we introduce

| | |
|---|---|
| latent state posterior | $q_\phi(s_t \mid s_{t-1}, a_{t-1}, o_t)$ |
| and latent action posterior | $q_\phi(u_t \mid s_t, a_t)$ |

as inference models. The latent state posterior encodes observations $o_t$ to latent states $s_t$ by using the deterministic transition. The latent action posterior encodes actions $a_t$ to latent actions $u_t$ conditioned on latent states $s_t$. All parameters of the generative model are indicated by $\theta$ and parameters of the inference model by $\phi$.

We derive the ELBO

$$
\log p(o_{1:T}, a_{1:T}) \geq \sum_{t=1}^{T} \Big[ \mathbb{E}_{s_t, u_t \sim q_\phi} \big[ \underbrace{\log p_\theta(o_t \mid s_t)}_{\text{observation reconstruction}} + \underbrace{\log p_\theta(a_t \mid s_t, u_t)}_{\text{action reconstruction}} \big]
$$
$$
- \underbrace{\text{KL}\big(q_\phi(s_t \mid s_{t-1}, a_{t-1}, o_t) \,\|\, p_\theta(s_t \mid s_{t-1}, u_{t-1})\big)}_{\text{observation consistency}} \tag{4}
$$
$$
- \underbrace{\text{KL}\big(q_\phi(u_t \mid s_t, a_t) \,\|\, p_\theta(u_t \mid s_t,)\big)}_{\text{action consistency}} \Big] =: -L_{ELBO}(o_{1:T}, a_{1:T}),
$$

which can be organized into individual terms for reconstruction and consistency of actions and observations. The derivation can be found in Appendix A.

Maximizing the objective enables us to learn a model which can generate trajectories close to the data distribution $\mathcal{D}$ by sampling from both priors. As we want to use the model to learn a policy via latent imagination, we add a reward $p_\theta(r_t \mid s_t)$ and termination $p_\theta(t_t \mid s_t)$ model. Hence, the complete model training objective is

$$L(o_{1:T}, a_{1:T}) = L_{ELBO}(o_{1:T}, a_{1:T}) - \sum_{t=1}^{T} \mathbb{E}_{s_t, u_t \sim q_\phi}[log(p_\theta(r_t \mid s_t)) + log(p_\theta(t_t \mid s_t))]. \quad (5)$$

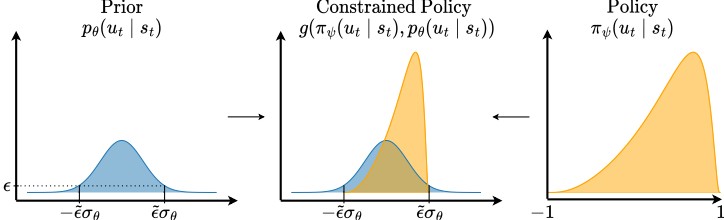

Figure 3: Policy constraint through explicit parametrization by using a linear transformation $g$ of the latent action prior $p_\theta(u_t \mid s_t)$ and the bounded policy $\pi_\psi(u_t \mid s_t) \in [-1, 1]$. The generated actions $a_t \sim p(a_t \mid s_t, g(\pi_\psi(u_t \mid s_t), p_\theta(u_t \mid s_t))$ are implicitly constrained to the data distribution.

**Constrained latent action policy**   We use the sequence model to generate imagined trajectories and use an actor-critic approach to train the policy. To predict state values we learn a value model $v_\xi(s_t)$ alongside the policy. Therefore we use the n-step return of a state

$$V_N^k(s_t) = \mathbb{E}_{s_t \sim p_\theta, u_t \sim \pi_\psi} \left[ \sum_{n=\tau}^{h-1} \lambda^{n-\tau} r_n + \lambda^{h-\tau} v_\xi(s_h) \right] \quad \text{with} \quad h = \min(\tau + k, t + H) \quad (6)$$

as regression target for $v_\xi(s_t)$ [27, 29]. Polices trained on trajectories generated by a model are prone to end up with degrading performance if the model only has access to a limited data distribution, as in the case of offline reinforcement learning. Compounding modeling errors and value overestimation of edge-of-reach states [19] are reasons for the decline. Since we train a generative action model, generated actions are implicitly constrained to the datasets action distribution by sampling from the action decoder $a_t \sim p_\theta(a_t \mid s_t, u_t)$. Hence, states outside the datasets observation distribution are hard to reach and our approach is resilient to value overestimation of edge-of-reach states. Compounding modeling errors are still a source of diminishing performance, but can be counteracted by increasing the representation power of the model or generating only short trajectories. To leverage the generative action model, we learn a policy $\pi_\psi(u_t \mid s_t)$ in the latent action space similar to [10, 25]. But, as both, the latent action prior $p_\theta(u_t \mid s_t)$ and the policy $\pi_\psi(u_t \mid s_t)$ are flexible, it is not ensured that they share the same support. Thus, we formulate policy optimization as a constrained optimization problem

$$\max_\psi \mathbb{E}_{s_t \sim p_\theta, \hat{u}_t \sim \pi_\psi} \left[ \sum_{\tau=t}^{t+H} V_N^k(s_\tau) \right] \quad (7)$$

$$\text{s.t. } \mathbb{E}_{s_t \sim p_\theta, \hat{u}_t \sim \pi_\psi} \left[ p_\theta(\hat{u}_t \mid s_t) \right] \geq \epsilon$$

similar to a support constraint [9]. We implement the constraint explicitly through parametrization to stay within the support, but do not impose any restrictions inside the supported limits (Figure 3). This is different to using a divergence measure which on one hand does not strictly ensure support limits and on the other hand is more restrictive as it also imposes a constraint on the shape of a distribution. Here and in the following $\hat{u}_t$ stands for a latent action sampled from the policy $\pi_\psi(u_t \mid s_t)$.

The policy is trained to maximize the n-step return $V_N^k(s_\tau)$ while staying in support of the latent action prior. Since the latent action prior $p_\theta(u_t \mid s_t)$ is normally distributed as $\mathcal{N}(\mu_\theta(s_t), \sigma_\theta(s_t))$, we can express the constraint as

$$p_\theta(\hat{u}_t \mid s_t) \geq \epsilon = p_\theta(\mu_\theta + \tilde{\epsilon}\sigma_\theta \mid s_t) \quad (8)$$

with $\tilde{\epsilon}$ as a parameter setting support as multiples of $\sigma_\theta$ centered around $\mu_\theta$. From the properties of a normal distributed variable follows that

$$\mu_\theta + \tilde{\epsilon}\sigma_\theta \geq \hat{u}_t \geq \mu_\theta - \tilde{\epsilon}\sigma_\theta. \tag{9}$$

We implement the constraint explicitly by parameterizing the policy as a linear function $g$ dependent on $\pi_\psi(u_t \mid s_t)$ and $p_\theta(u_t \mid s_t)$:

$$\mu_\theta + \tilde{\epsilon}\sigma_\theta \geq g(\pi_\psi(u_t \mid s_t), p_\theta(u_t \mid s_t)) \geq \mu_\theta - \tilde{\epsilon}\sigma_\theta. \tag{10}$$

The support of the policy distribution is chosen to be bounded

$$\hat{u}_t \sim \pi_\psi(u_t \mid s_t), \quad \hat{u}_t \in [-1, 1] \tag{11}$$

and $g(\pi_\psi(u_t \mid s_t), p_\theta(u_t \mid s_t))$ as a linear combination of the latent action predicted by the policy $\hat{u}_t$ and the distribution parameters $\sigma_\theta$ and $\mu_\theta$ of the latent action prior:

$$g(\hat{u}_t, \mu_\theta, \sigma_\theta) = \mu_\theta + \hat{u}_t \cdot \tilde{\epsilon} \cdot \sigma_\theta. \tag{12}$$

## 4 Experiments

In the next section, we evaluate the effectiveness of our approach. It is divided into three parts: first, we assess the performance using standard benchmarks; then, we study how different design choices affect value overestimation; lastly, we analyze the influence of the support constraint parameter. We additionally provide the final performances in two tables in Appendix E.

We limit our benchmark evaluation to the most relevant state-of-the-art offline reinforcement learning methods to answer the following questions: 1) How do latent action state-space models compare to state-space models? 2) How comparable are model-free methods focusing on latent action spaces to latent action state-space models? 3) Does C-LAP suffer from value overestimation? 4) How does the support constraint affect the performance? 5) How does the performance differ between visual observations and observations with low-dimensional features? To focus on the latter, we separately evaluate the performance on low-dimensional feature observations using the D4RL benchmark [26], and on image observations using the V-D4RL benchmark [20].

### 4.1 Benchmark results

**D4RL**  Since most offline model-based reinforcement learning methods are designed for observations with low-dimensional feature observations, there exist many options for comparison. We make a selection to include the most relevant methods focusing on latent actions and auto-regressive model-based reinforcement learning. Therefore, we include the following methods: PLAS, which is a model-free method using a latent action space [10]. MOPO, a probabilistic ensemble-based offline model-based reinforcement learning method using a modification to the Bellman update to penalize high variance in next state predictions [16]. And MOBILE, which is similar to MOPO, but penalizes high variance in value estimates instead [18]. We compare the algorithms on three different locomotion environments, namely halfcheetah, walker2d and hopper, with four datasets (medium-replay, medium, medium-expert, expert) each and the antmaze navigation environment with four datasets (umaze, umaze-diverse, medium-play, medium-diverse). The results, shown in Figure 4, display the mean and standard deviation of normalized returns over four seeds during the phase of policy training, with steps denoting gradient steps. The dashed lines indicate the asymptotic performance for MOPO and MOBILE. A detailed summary of all implementation details is provided in the Appendix D.

When comparing C-LAP to PLAS, we find that learning a joint generative model of actions and observations outperforms a generative model of only actions when used with actor-critic reinforcement learning. Both methods can use the generative nature of the model to speed up policy learning, which becomes especially clear in the results on all locomotion expert and medium-expert datasets. Compared to MOPO and MOBILE, C-LAP shows a superior or comparable performance on all datasets except halfcheetah-medium-replay-v2, halfcheetah-medium-v2 and hopper-medium-v2. Especially outperforming on the antmaze environment, where MOPO and MOBILE fail to solve the task for any of the considered datasets. The asymptotic performance of MOBILE on locomotion environments sometimes exceeds the results of C-LAP, but needs three times as many gradient steps.

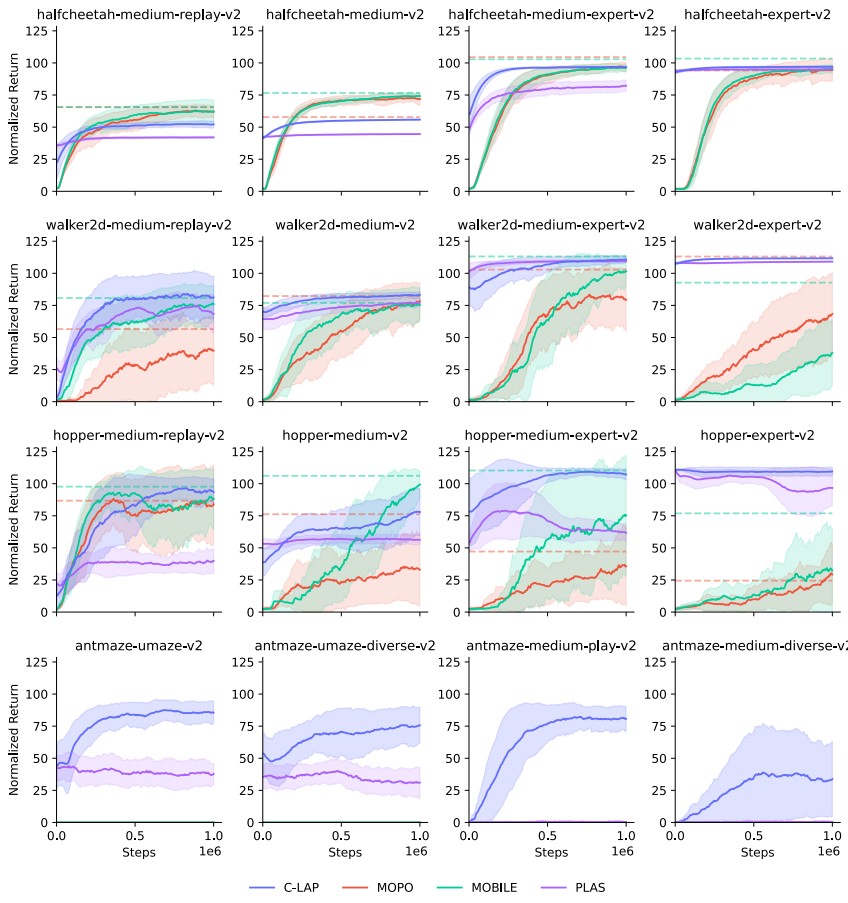

Figure 4: Evaluation on low-dimensional feature observations using D4RL benchmark datasets. We plot mean and standard deviation of normalized returns over 4 seeds.

Overall the results indicate that latent action state-space models with constrained latent action polices not only match the state-of-the-art on observations with low-dimensional features as observations, but also jump-start policy learning by using the action decoder to sample actions that lead to high rewards already after the first gradient steps: If the dataset is narrow, generated actions when sampling from the latent action prior will fall into the same narrow distribution. For instance, in an expert dataset, sampled actions will also be expert-level actions. During policy training, instead of sampling from this prior, we restrict the support of the policy dependent on the latent action prior. Thus, sampled latent actions from the policy will always be decoded to fall into the dataset's action distribution. So even a randomly initialized policy in the beginning of the training can generate a high reward by using the latent action decoder. This effect is especially prominent in narrow datasets such as expert datasets.

**V-D4RL**   There are currently few auto-regressive model-based reinforcement learning methods that specifically target visual observations, with none emphasizing latent actions. In our evaluation, we include LOMPO [21] and Offline DV2 [20]. Both methods use a latent state space model and an uncertainty penalized reward. However the specifics of the penalty calculations are different: while LOMPO uses standard deviation of log probabilities as penalty, Offline DV2 uses mean disagreement. Additionally, LOMPO trains an agent on a mix of real and imagined trajectories with an off-policy actor-critic approach, whereas Offline DV2 exclusively trains on imagined trajectories and back-propagates gradients through the dynamics model. Further implementation details are included in Appendix D.

C-LAP demonstrates superior performance across all datasets, especially significant on cheetah-run-medium_expert, walker-walk-medium_expert and walker-walk_expert. Datasets with a large

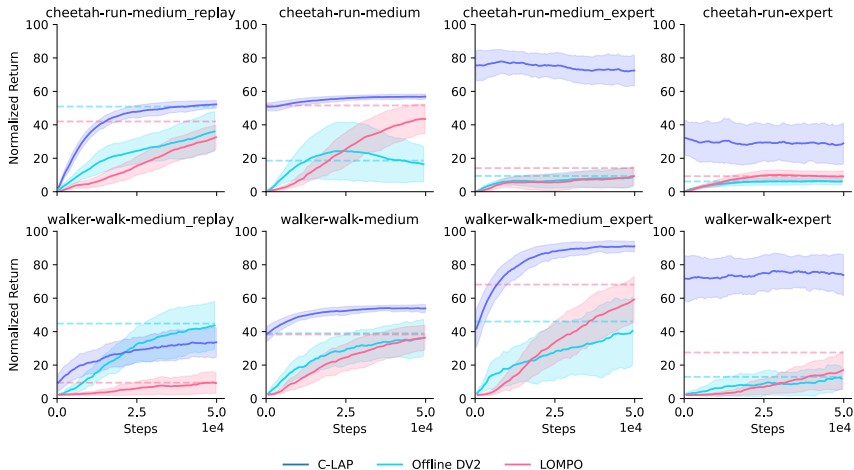

Figure 5: Evaluation on visual observations using V-D4RL benchmark datasets. We plot mean and standard deviation of normalized returns over 4 seeds.

diversity of actions, such as medium-replay datasets, exhibit a weaker inductive bias for a generative action model. Hence, they require more additional policy steps, as can be seen for both the D4RL and V-D4RL benchmarks.

## 4.2 Value overestimation

Limiting value overestimation plays a central role in offline reinforcement learning. To evaluate the effectiveness of C-LAP, we report value estimates alongside normalized returns on all walker2d datasets in Figure 6. A similar analysis for all considered baselines is provided in Appendix G. To further analyze the influence of different action space design choices, we include the following ablations: a variant *no constraint*, which does not formulate policy optimization as constrained objective, but uses a Gaussian policy distribution to potentially cover the whole Gaussian latent action space; and a variant *no latent action*, which does not emphasize latent actions, but uses a regular state-space model as in Dreamer [29]. Besides that, we added dashed lines to indicate the dataset's average return and average maximum value estimate. The *no latent action* variant fails to learn

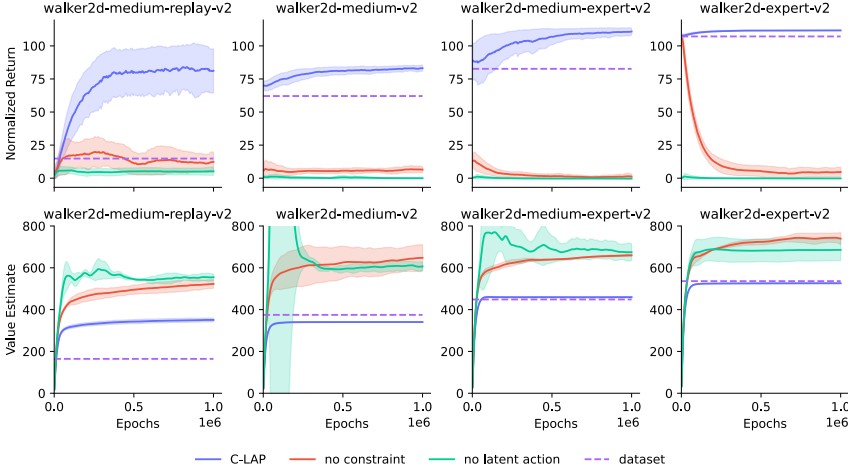

Figure 6: Ablation study, comparing C-LAP to the following variants: no constraint, C-LAP without enforcing the policy constraint dependent on the action prior; no latent action, C-LAP without a latent action space similar to Dreamer [29]. We plot mean and standard deviation of normalized returns and value estimates over 3 seeds. Moreover we add the dataset's average return and average maximum value estimate indicated by dashed lines.

an effective policy: normalized returns are almost zero and the dataset's reference returns remain unattained; value estimates are significantly exceeding the dataset's reference values, indicating value overestimation. The *no constraint* variant can use the generative action decoder to limit generated actions to the dataset's action distribution, but the Gaussian policy is free to move to regions which are unlikely under the action prior. Thus, nullifying the implicit constraint imposed by the action decoder, resulting in collapsing returns and value overestimation. Only C-LAP achieves a high return and generates value estimates which are close to the dataset's reference. The value estimates on walker2d-medium-replay-v2 are higher than the dataset's reference, as the agent's performance is also exceeding the reference performance. The results confirm the importance of limiting value overestimation in offline reinforcement learning, and demonstrate that constraining latent action policies can be an effective measure for achieving this.

### 4.3   Support constraint parameter

To evaluate the influence of the support constraint parameter $\tilde{\epsilon}$ on the performance of C-LAP, we perform a sensitivity analysis across all walker2d datasets (Figure 7). Except for the more diverse medium-replay-v2 dataset, adjusting $\tilde{\epsilon}$ from $0.5$ to $3.0$ only has a minor impact on the achieved return. However, when choosing an unreasonable large value such as $\tilde{\epsilon} = 10.0$ or removing the constraint altogether (Figure 6), we observe a collapse during training. This highlights a key insight: constraining the policy to the support of the latent action prior is essential. And in many cases, using a smaller support region closer to the mean (small $\tilde{\epsilon}$) proves sufficient.

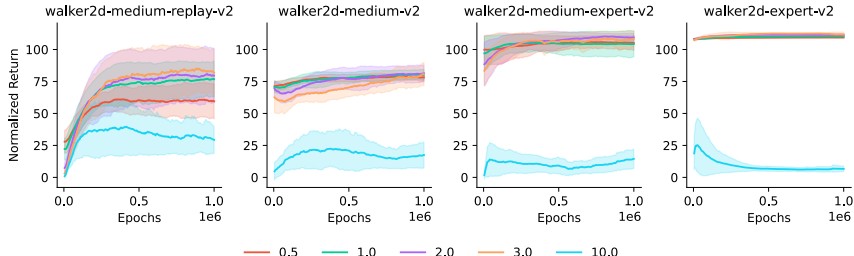

Figure 7: Sensitivity analysis of the support constraint parameter $\tilde{\epsilon}$ for the considered D4RL walker2d datasets. We plot mean and standard deviation of normalized returns over 4 seeds.

## 5   Related Work

Offline reinforcement learning methods fall into two groups: model-free and model-based. Both types aim to tackle problems like distribution shift and value overestimation. This happens because the methods use a fixed dataset rather than learning by interacting with the environment.

**Model-free**   Current model-free methods typically work by limiting the learned policy or by regularizing value estimates. TD3+BC [11] adds a behavior cloning regularization term to the policy update objective to enforce actions generated by the policy to be close to the dataset's action distribution. Similarly, SPOT [9] includes a regularization term in the policy update, derived from a support constraint perspective. It also uses a conditional variational auto-encoder (CVAE) to estimate the behavior distribution. Following a comparable intention, BEAR [8] constraints the policy to the support of the behavior policy via maximum mean discrepancy. BCQ [7] and PLAS [10] use a CVAE similarly but don't use a regularization term. Instead, they constrain the policy implicitly by making the generative model part of the policy. Beside these methods, many other approaches exist, with CQL [12] and IQL [14] being some of the most well-known. CQL uses a conservative policy update by setting a lower bound on value estimates to prevent overestimation, while IQL avoids out-of-distribution values by using a modified SARSA-like objective in combination with expectile regression to only use state-action tuples contained in the dataset.

**Model-based**   Model-based offline reinforcement learning methods learn a dynamics model to generate samples for policy training. This basically converts offline learning to an online learning problem. Model-based methods mainly address model errors and value overestimation by using

a probabilistic ensemble and adding an uncertainty penalty in the Bellman update. MOPO [16] uses a probabilistic ensemble as in [32] and adds the maximum standard deviation of all ensemble predictions as uncertainty penalty. Similar to that, MOReL [17] adheres to the same methodology, but uses pairwise maximum difference of the ensemble predictions as penalty instead. Analogously, MOBILE [18] estimates the values for all by the ensemble predicted states and uses the standard deviation of value estimates as penalty. Edge of reach [19] comes to the conclusion that value estimation on edge of reach states are the overarching issue compared to model errors. In the end, they come up with a comparable solution to MOBILE, but use an ensemble of value networks alongside the ensemble of dynamic models. COMBO [28] pursues a different approach, as they integrate the conservatism of CQL into value function updates, removing the need for uncertainty penalties. Besides that, some methods use a different class of models: instead of learning a predictive model in observation space, they use latent state-space models to make predictions on latent states. Among these methods is LOMPO [21], which builds up on Dreamer [29], but integrates an ensemble to predict stochastic states and use the standard deviation of the log probability of the ensemble predictions as an uncertainty penalty similar to previous methods. The policy is trained on a mix of imagined and real world samples, hence they use an off-policy actor-critic style approach for policy learning. Offline DV2 [20] uses a similar model, but is based on a different penalty. Namely, they use the difference between the individual ensemble mean predictions and mean over all ensembles as uncertainty penalty. Furthermore, the policy is trained only on imagined trajectories with gradients calculated by back-propagating through the dynamics model. Overall, Offline DV2 is the method most comparable to our approach, but still different in many ways as we propose a latent action state-space model compared to a usual state-space model, and frame policy learning as constrained optimization. So far all discussed models operate in an auto-regressive fashion, but another class of methods exists, which casts offline model-based reinforcement learning as trajectory modeling. Instead of learning a policy, these kind of approaches integrate decision making and modeling of the underlying dynamics into a single objective and use the model for planning. Among them are Diffuser [24], which employs guided diffusion for planning; TT [22], which builds on advances in transformers; and TAP [23], which uses a VQ-VAE with a transformer-based architecture to create a discrete latent action space for planning.

# 6   Conclusion

We present C-LAP, a model-based offline reinforcement learning method. To tackle the issue of value overestimation, we first propose an auto-regressive latent-action state space model to learn a generative model of the joint distribution of observations and actions. Second, we propose a method for policy training to stay within the dataset's action distribution. We explicitly parameterize the policy depending on the latent action prior and formulate policy learning as constrained objective similar to a support constraint. We find that C-LAP significantly speeds-up policy learning, is competitive on the D4RL benchmark and especially outperforms on the V-D4RL benchmark, raising the best average score across all dataset's from previously $31.5$ to $58.8$.

**Limitations**   Depending on the dataset and environment the effectiveness of C-LAP differs: Datasets which only contain random actions are challenging for learning a generative action model, thus we do not include them in our evaluation. The effect of jump-starting policy learning with the latent action decoder to already achieve high rewards in the beginning of policy training is prominent in narrow datasets, but less effective for diverse datasets. While training the model of C-LAP does not require additional gradient steps, it still takes more time compared to LOMPO [21] and Offline DV2 [20] as the latent action state-space model is more complex than a usual latent state-space model.

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

# A  ELBO derivation

We start with Equation (1) to jointly model the dataset's observation and action distribution

$$\log p(o_{1:T}, a_{1:T}) = \log \int p(o_{1:T}, a_{1:T} \mid s_{1:T}, u_{1:T}) p(s_{1:T}, u_{1:T}) \, ds \, du, \tag{13}$$

and the defintions in Equation (3), namely:

$$p(o_{1:T}, a_{1:T} \mid s_{1:T}, u_{1:T}) = \prod_{t=1}^{T} p(o_t \mid s_t) p(a_t \mid s_t, u_t), \tag{14}$$

$$p(s_{1:T}, u_{1:T}) = \prod_{t=1}^{T} p(u_t \mid s_t) p(s_t \mid s_{t-1}, u_{t-1}). \tag{15}$$

We introduce the inference model

$$q(s_t, u_t \mid s_{t-1}, a_{t-1}, a_t, o_t) = q(s_t \mid s_{t-1}, a_{t-1}, o_t) q(u_t \mid s_t, a_t) \tag{16}$$

and by replacing $p(o_{1:T}, a_{1:T} \mid s_{1:T}, u_{1:T})$ and $p(s_{1:T}, u_{1:T})$ in Equation (13) we get:

$$\log p(o_{1:T}, a_{1:T}) = \log \int \prod_{t=1}^{T} p(o_t \mid s_t) p(a_t \mid s_t, u_t) p(u_t \mid s_t) p(s_t \mid s_{t-1}, u_{t-1}) \, ds \, du. \tag{17}$$

We include the inference model and resolve using Jensen's Inequality:

$$\log p(o_{1:T}, a_{1:T}) = \log \int \prod_{t=1}^{T} p(o_t \mid s_t) p(a_t \mid s_t, u_t) p(u_t \mid s_t) p(s_t \mid s_{t-1}, u_{t-1}) \frac{q(s_t, u_t \mid s_{t-1}, a_{t-1}, a_t, o_t)}{q(s_t, u_t \mid s_{t-1}, a_{t-1}, a_t, o_t)} \, ds \, du \tag{18}$$

$$= \log \mathbb{E}_q [\prod_{t=1}^{T} \frac{p(o_t \mid s_t) p(a_t \mid s_t, u_t) p(u_t \mid s_t) p(s_t \mid s_{t-1}, u_{t-1})}{q(s_t, u_t \mid s_{t-1}, a_{t-1}, a_t, o_t)}] \tag{19}$$

$$= \log \mathbb{E}_q [\prod_{t=1}^{T} p(o_t \mid s_t) p(a_t \mid s_t, u_t) \frac{p(s_t \mid s_{t-1}, u_{t-1})}{q(s_t \mid s_{t-1}, a_{t-1}, o_t)} \frac{p(u_t \mid s_t)}{q(u_t \mid s_t, a_t)}] \tag{20}$$

$$\geq \mathbb{E}_q [\sum_{t=1}^{T} \log p(o_t \mid s_t) p(a_t \mid s_t, u_t) + \log \frac{p(s_t \mid s_{t-1}, u_{t-1})}{q(s_t \mid s_{t-1}, a_{t-1}, o_t)} + \log \frac{p(u_t \mid s_t)}{q(u_t \mid s_t, a_t)}] \tag{21}$$

$$= \sum_{t=1}^{T} \big[ \mathbb{E}_q [\log p(o_t \mid s_t) p(a_t \mid s_t, u_t)] \tag{22}$$

$$- \mathrm{KL}(q(s_t \mid s_{t-1}, a_{t-1}, o_t) \, || \, p(s_t \mid s_{t-1}, u_{t-1}) \tag{23}$$

$$- \mathrm{KL}(q(u_t \mid s_t, a_t) \, || \, p_\theta(u_t \mid s_t)] \big] \tag{24}$$

$$=: -L_{ELBO}(o_{1:T}, a_{1:T}). \tag{25}$$

## B  Algorithm

We provide the general algorithm of C-LAP.

---
**Algorithm 1:** C-LAP

---
Given: Dataset $\mathcal{D}$, learning rates $\alpha_\theta, \alpha_\phi, \alpha_\psi, \alpha_\xi$, sequence length $K$, dream rollout length $H$

`// Model Training`

Initialize model parameters $\theta$ and $\phi$
**for** *epoch in model training epochs* **do**
    **for** *update step t = 1..T* **do**
        Sample batch of trajectories $\{(o_t, a_t, r_t, t_t)\}_{t=k}^{k+K} \sim \mathcal{D}$
        Compute model training objective $L(o_{1:T}, a_{1:T})$ via Equation (5)
        Update model parameters $\theta \leftarrow \theta + \alpha_\theta \nabla_\theta L(o_{1:T}, a_{1:T})$, $\phi \leftarrow \phi + \alpha_\phi \nabla_\phi L(o_{1:T}, a_{1:T})$
    **end for**
**end for**

`// Agent Training`

Initialize policy and critic parameters $\psi$ and $\xi$
**for** *epoch in agent training epochs* **do**
    **for** *update step t = 1..T* **do**
        Sample batch of trajectories $\{(o_t, a_t, r_t, t_t)\}_{t=k}^{k+L} \sim \mathcal{D}$
        Compute latent states $s_t \sim q_\phi(s_t \mid s_{t-1}, a_{t-1}, o_t)$
        Sample random starting state $s_{init}$ from each trajectory
        Create dream rollouts $\{(s_\tau, u_\tau, a_\tau)\}_{\tau=t}^{t+H}$ from $s_{init}$ using $p_\theta(s_t \mid s_{t-1}, u_{t-1})$ and
         $a_t \sim p(a_t \mid s_t, g(\pi_\psi(u_t \mid s_t), p_\theta(u_t \mid s_t))$
        Predict rewards $r_\tau \sim p_\theta(r_\tau \mid s_\tau)$, termination $t_\tau \sim p_\theta(t_\tau \mid s_\tau)$ and values
         $v_\xi \sim v_\psi(u_\tau \mid s_\tau)$
        Compute value estimates $V_N^k(s_\tau)$ via Equation (6)
        Update policy parameters $\psi \leftarrow \psi + \alpha_\psi \nabla_\psi \sum_{\tau=t}^{t+H} V_N^k(s_\tau)$
        Update critic parameters $\xi \leftarrow \xi + \alpha_\xi \nabla_\theta \sum_{\tau=t}^{t+H} \frac{1}{2} \|v_\xi(s_\tau) - V_N^k(s_\tau)\|$
    **end for**
**end for**

---

## C  Computational Resources

We use a different hardware setup for experiments with visual observations and experiments with low-dimensional features as observations. We run all experiments on a shared local cluster. C-LAP experiments with visual observations take around 10 hours on a RTX8000 GPU and experiments with low-dimension feature observations around 11 hours on a A100 GPU. We aim to execute most of our code on GPU and parallelize our implementation. Environment evaluations represent a bottleneck as they require execution on CPU. Overall, it takes around 70 combined GPU days to reproduce the benchmark results of all methods. This does not include the compute required for the evaluation of preliminary implementations or hyper-parameter settings.

## D  Implementation Details

We implement all methods in JAX [33] using Equinox [34]. We provide the hyper-parameters of C-LAP in Table 2 and the constraint values used for the D4RL benchmark in Table 3 and for the V-D4RL benchmark in Table 4. In general we consider constraint values in the range $\tilde{\epsilon} \in \{0.5, 1.0, 2.0, 3.0\}$.

We implement MOPO and MOBILE following [18] and use the respective hyper-parameters provided in the publication. For MOPO, we select the max-aleatoric version. As no hyper-parameters are provided for the expert datasets of the D4RL benchmark we sweep through the range specified in Table 1 and use the one selected in the table. For the antmaze environment we additionally evaluate

the discount factors of $\{0.99, 0.995, 0.997\}$ to compensate for the sparse reward, but do not find hyper-parameters to solve the task.

We implement PLAS following [10] and use the default hyper-parameters for the expert datasets.

We implement LOMPO and Offline DV2 using the continuous state implementation from Dreamer [29], which is different from the original implementation using discrete states. This change keeps the model closer to C-LAP, which is also using continuous states. For LOMPO and Offline DV2, we take hyper-parameters from [20].

As all our implementations might differ from the original one, we include the original scores from the paper alongside our results in Table 5 and Table 6.

| | MOPO | MOBILE |
|---|---|---|
| Hyper-parameter range | penalty coefficient $\in \{0.5, 2.5, 5.0, 10.0\}$
rollout steps: $\in \{1, 5\}$
dataset ratio: $\in \{0.05, 0.5, 0.8\}$ | penalty coefficient $\in \{1.5, 2.5, 3.5\}$
rollout steps: $\in \{1, 5\}$
dataset ratio: $\in \{0.05, 0.5, 0.8\}$ |
| halfcheetah-expert-v2 | penalty coefficient: 1.0
rollout steps: 5
dataset ratio: 0.5 | penalty coefficient: 2.5
rollout steps: 5
dataset ratio: 0.5 |
| walker2d-expert-v2 | penalty coefficient: 10.0
rollout steps: 1
dataset ratio: 0.8 | penalty coefficient: 2.5
rollout steps: 1
dataset ratio: 0.5 |
| hopper-expert-v2 | penalty coefficient: 5.0
rollout steps: 5
dataset ratio: 0.05 | penalty coefficient: 3.5
rollout steps: 5
dataset ratio: 0.5 |

Table 1: MOPO and MOBILE hyper-parameters for the expert datasets in the D4RL benchmark.

| Environment | Dataset | Constraint $\bar{\epsilon}$ |
|---|---|---|
| Halfcheetah | medium-replay | 3.0 |
| | medium | 3.0 |
| | medium-expert | 2.0 |
| | expert | 2.0 |
| Walker2d | medium-replay | 3.0 |
| | medium | 1.0 |
| | medium-expert | 3.0 |
| | expert | 2.0 |
| Hopper | medium-replay | 2.0 |
| | medium | 3.0 |
| | medium-expert | 2.0 |
| | expert | 0.5 |
| Antmaze | umaze | 1.0 |
| | umaze-diverse | 1.0 |
| | medium-play | 1.0 |
| | mediun-diverse | 1.0 |

Table 3: C-LAP constraint values for the D4RL benchmark

| Environment | Dataset | Constraint $\bar{\epsilon}$ |
|---|---|---|
| Cheetah-run | medium-replay | 3.0 |
| | medium | 3.0 |
| | medium-expert | 3.0 |
| | expert | 3.0 |
| Walker-walk | medium-replay | 3.0 |
| | medium | 3.0 |
| | medium-expert | 0.5 |
| | expert | 0.5 |

Table 4: C-LAP constraint values for the V-D4RL benchmark

| Model | |
|---|---|
| Stochastic latent state size | 30
64 (antmaze) |
| Deterministic latent state size | 200 (low-dimensional features)
512 (visual, antmaze) |
| Observation embedding size | 30 (low-dimensional features)
64 (antmaze)
1024 (visual) |
| Latent action size | 12 |
| Hidden units | 200 |
| Hidden activation | selu |
| State encoder | MLP
units: 128
layers: 2

CNN
channels: $[32, 64, 128, 256]$
kernels: $[4, 4, 4, 4]$
stride: 2 |
| State decoder | MLP
units: 128
layers: 2
distribution: Gaussian

CNN
channels: $[128, 64, 32, 3]$
kernels: $[5, 5, 6, 6]$
stride: 2
distribution: Gaussian |
| Latent action encoder | units: 512
layers: 2
distribution: Gaussian |
| Latent action decoder | units: 512
layers: 2
distribution: Beta |
| Latent action prior | units: 256
layers: 2
distribution: Gaussian |
| Learning rate | $3e-4$ |
| Batch size | 64 |
| Window length | 50 |
| Agent | |
| Hidden activation | selu |
| Policy | units: 256
layers: 3
distribution: TanhGaussian |
| Value | units: 256
layers: 3
number of networks: 2 |
| Learning rate | $8e-5$ |
| Logprob/entropy regulariser scaling | 0.01 |
| Dream rollout length | 5
10 (antmaze) |
| Discount factor | 0.99
0.997 (antmaze-umaze, antmaze-medium-diverse)
0.998 (antmaze-umaze-diverse, antmaze-medium-play) |

Table 2: C-LAP hyper-parameters

# E D4RL and V-D4RL benchmark results

We provide the benchmark results in the usual offline reinforcement learning table format. As our implementations might differ from the original one, we include the original scores from the paper alongside our results in tables. We also include the average over all datasets except the expert datasets to make it comparable to the average scores provided in the respective publication.

| Environment | Dataset | PLAS (paper) [10] | PLAS | MOPO (paper) [18] | MOPO | MOBILE (paper) [18] | MOBILE | C-LAP |
|---|---|---|---|---|---|---|---|---|
| Halfcheetah | medium-replay | 43.9 | $41.8 \pm 0.7$ | 72.1 | $66.3 \pm 5.3$ | $71.7 \pm 1.2$ | $63.7 \pm 6.2$ | $55.5 \pm 4.5$ |
| | medium | 39.4 | $44.8 \pm 0.5$ | 72.4 | $57.5 \pm 39.4$ | $74.6 \pm 1.2$ | $77.3 \pm 2.0$ | $56.2 \pm 4.8$ |
| | medium-expert | 96.6 | $82.5 \pm 5.5$ | 83.6 | $104.4 \pm 2.7$ | $108.2 \pm 2.5$ | $103.2 \pm 0.7$ | $96.8 \pm 0.3$ |
| | expert | - | $94.8 \pm 1.5$ | - | $94.0 \pm 25.7$ | - | $103.0 \pm 1.0$ | $97.1 \pm 0.6$ |
| Walker2d | medium-replay | 30.2 | $67.7 \pm 14.9$ | 85.2 | $54.6 \pm 20.5$ | $89.9 \pm 1.5$ | $85.4 \pm 10.6$ | $86.0 \pm 20.1$ |
| | medium | 44.6 | $79.4 \pm 1.8$ | 84.1 | $78.0 \pm 22.5$ | $87.7 \pm 1.1$ | $81.0 \pm 0.5$ | $82.5 \pm 4.4$ |
| | medium-expert | 89.6 | $109.4 \pm 1.0$ | 105.3 | $105.6 \pm 7.1$ | $115.2 \pm 0.7$ | $113.5 \pm 4.0$ | $111.8 \pm 1.0$ |
| | expert | - | $109.1 \pm 0.4$ | - | $113.4 \pm 0.5$ | - | $15.0 \pm 17.3$ | $111.7 \pm 0.3$ |
| Hopper | medium-replay | 27.9 | $49.1 \pm 15.1$ | 92.8 | $87.0 \pm 36.2$ | $103.9 \pm 1.0$ | $96.1 \pm 18.2$ | $78.6 \pm 29.9$ |
| | medium | 32.9 | $57.0 \pm 4.3$ | 62.8 | $76.8 \pm 41.3$ | $106.6 \pm 0.6$ | $106.2 \pm 0.2$ | $80.3 \pm 18.6$ |
| | medium-expert | 111.0 | $55.7 \pm 6.4$ | 74.9 | $43.5 \pm 28.2$ | $112.6 \pm 0.2$ | $111.8 \pm 1.8$ | $105.0 \pm 7.7$ |
| | expert | - | $97.1 \pm 16.3$ | - | $26.2 \pm 5.1$ | - | $78.6 \pm 38.4$ | $110.5 \pm 3.4$ |
| **Locomotion average without expert** | | 57.3 | 65.3 | 81.5 | 74.9 | 96.7 | 93.1 | 83.6 |
| **Locomotion average** | | - | 74.0 | - | 75.6 | - | 92.7 | 89.3 |
| Antmaze | umaze | 0.7 | $46.7 \pm 7.6$ | - | 0.0 | - | 0.0 | $81.3 \pm 4.8$ |
| | umaze-diverse | 0.5 | $33.3 \pm 14.4$ | - | 0.0 | - | 0.0 | $75.0 \pm 20.4$ |
| | medium-play | 0.2 | 0.0 | - | 0.0 | - | 0.0 | $77.5 \pm 9.6$ |
| | medium-diverse | 0.0 | 0.0 | - | 0.0 | - | 0.0 | $45.0 \pm 37.0$ |
| **Navigation average** | | 0.35 | 20.0 | - | 0.0 | - | 0.0 | 69.7 |

Table 5: Results on the D4RL benchmark. Showing normalized returns and standard deviations at the end of policy training.

| Environment | Dataset | Offline DV2 (paper) [20] | Offline DV2 | LOMPO (paper) [20] | LOMPO | C-LAP |
|---|---|---|---|---|---|---|
| Cheetah-run | medium-replay | $61.6 \pm 1.0$ | $54.6 \pm 6.5$ | $36.3 \pm 13.6$ | $42.2 \pm 3.5$ | $52.5 \pm 1.3$ |
| | medium | $17.2 \pm 3.5$ | $16.8 \pm 8.7$ | $16.4 \pm 8.3$ | $52.3 \pm 3.8$ | $57.4 \pm 0.9$ |
| | medium-expert | $10.4 \pm 3.5$ | $9.4 \pm 6.3$ | $11.9 \pm 1.9$ | $14.9 \pm 5.9$ | $73.2 \pm 8.5$ |
| | expert | $10.9 \pm 3.2$ | $5.3 \pm 1.4$ | $14.0 \pm 3.8$ | $8.4 \pm 5.3$ | $36.6 \pm 11.3$ |
| Walker-walk | medium-replay | $56.6 \pm 18.1$ | $42.5 \pm 14.2$ | $34.7 \pm 19.7$ | $11.4 \pm 5.8$ | $34.5 \pm 5.5$ |
| | medium | $34.1 \pm 19.7$ | $40.6 \pm 10.8$ | $43.4 \pm 11.1$ | $39.0 \pm 5.7$ | $54.8 \pm 1.2$ |
| | medium-expert | $43.9 \pm 34.4$ | $41.0 \pm 20.0$ | $39.2 \pm 19.5$ | $58.0 \pm 5.3$ | $93.2 \pm 2.2$ |
| | expert | $4.8 \pm 0.6$ | $6.1 \pm 5.6$ | $5.3 \pm 7.7$ | $25.6 \pm 17.6$ | $68.1 \pm 14.0$ |
| **Average** | | 29.9 | 27.0 | 25.2 | 31.5 | 58.8 |

Table 6: Results on the V-D4RL benchmark. Showing normalized returns and standard deviations at the end of policy training.

# F Action distribution

To evaluate the claim that latent actions generated by the latent action prior are close to the dataset's action distribution we use the following approach: We randomly select one trajectory from the hopper-expert-v2 dataset and employ k-nearest neighbors to identify the 20 nearest observations and their corresponding actions within the whole dataset for each step. We then sample from the action prior and decoder to generate actions based on the nearest observations at each step. Thereafter, we split the selected trajectory into sections from leaving the ground to touching the ground and fit normal distributions to the aggregated dataset actions and reconstructed prior actions. Thus, we end up with an approximation of the dataset's action distribution and an approximation of the action distribution generated by the prior for each step in the trajectory. Figure 8 shows the aggregated distributions (from leaving the ground to touching the ground) for one exemplary trajectory of the hopper-expert-v2 dataset.

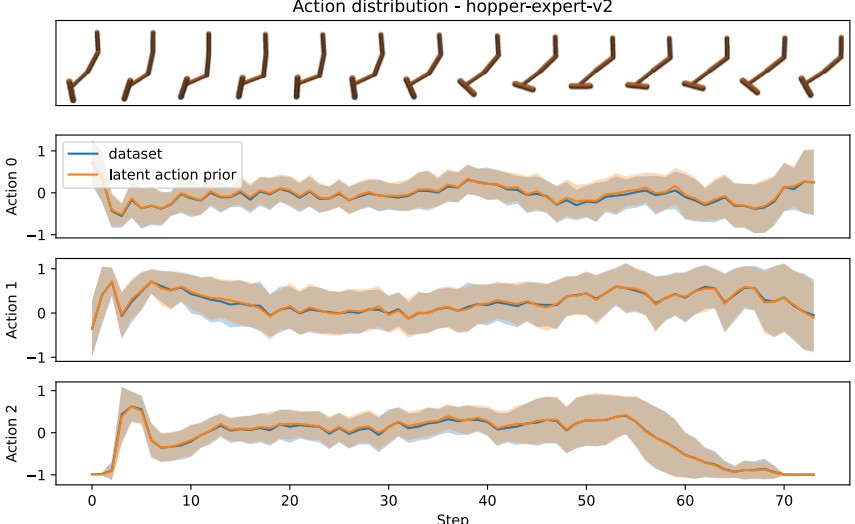

Figure 8: Comparison of the dataset's action distribution to the distribution of actions sampled from the latent action prior and latent action decoder. The figure shows the aggregated distributions (from leaving the ground to touching the ground) for one exemplary trajectory of the hopper-expert-v2 dataset

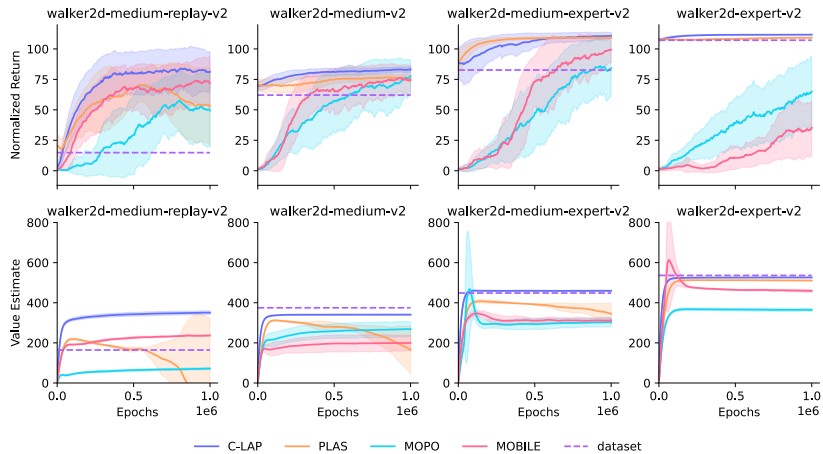

Figure 9: Evaluation of value overestimation for all baselines. We plot mean and standard deviation of normalized returns and value estimates over 3 seeds. Moreover, we add the dataset's average return and average maximum value estimate indicated by dashed lines.

## G Value overestimation for the considered baselines

We further report value estimates alongside normalized returns on all walker2d datasets for the considered baselines (Figure 9). As they estimate Q-values, we calculate the corresponding value estimates by averaging over 10 actions sampled from their respective policies. MOPO and MOBILE have a low value estimate, which can be attributed to the incorporated uncertainty penalties. PLAS's value estimates are only stable for walker2d-expert-v2 datasets, but collapse for the other considered datasets. Overall, it seems that value overestimation is not the cause for degrading performance for these methods.

