# OpenReview forum: "Constrained Latent Action Policies for Model-Based Offline Reinforcement Learning"
_NeurIPS.cc/2024/Conference — NeurIPS 2024 poster_

### Official Review · Reviewer_6cNB · 2024-07-10

**Soundness:** 2
**Presentation:** 3
**Contribution:** 3
**Rating:** 5
**Confidence:** 4

**Summary:**

This paper proposes C-LAP, a model-based offline RL method that mitigates the distributional shift problem without using uncertainty penalties or modifying the Bellman update. It learns a joint distribution of latent states and latent actions and constrains the latent action policy to the dataset distribution by using a linear transformation. Experiments on D4RL datasets and V-D4RL datasets report performance comparable to other strong offline RL baselines.

**Strengths:**

1. This paper addresses the issue of distributional shift in offline RL by incorporating latent actions to enhance policy learning. The empirical results demonstrate that the proposed method outperforms other offline RL baselines by requiring fewer gradient steps to learn a policy.
2. The idea of combining a latent action prior and a bounded policy to obtain the constrained policy is novel and intuitive.

**Weaknesses:**

1. Figure 1 requires a clearer presentation of the model diagram. It would be better to simultaneously show the prior and posterior in Figure 1(a), and the latent action prior $p_\theta(u_t|s_t)$ and the policy $\pi_\psi(u_t|s_t)$ in Figure 1(b) to improve understanding. During the policy training phase, are the imagined trajectories rolled out from the latent state prior $p_\theta(s_t|s_{t−1}, u_{t−1})$ or the latent state posterior $q_\phi(s_t|s_{t−1}, a_{t−1}, o_t)$? If it is the latent state prior, why is the output of the action decoder used? If it is the latent state posterior, $o_t$ is unavailable.
2. Could the authors provide some distribution results to support the claim that the latent actions generated by the action prior are close to the dataset's action distribution? Does the predictive power of using latent actions degrade in performance compared to models using real actions? Some quantitative or qualitative results can be provided.
3. It would be better to highlight the differences between the proposed method and Dreamer in terms of model designs and algorithms.
4. The experiments mainly focus on Mujoco tasks. How does the proposed method perform on navigation tasks such as AntMaze?
5. Why do some experiments in Figure 4, 5 show high returns at the beginning of policy learning (zero gradient step)?
6. In table 5 and table 6, the performance of C-LAP is inferior to some baseline models in medium or medium-relay datasets. This results seem to indicate that the effectiveness of the proposed method depends on the quality of the offline dataset.

**Questions:**

Please see the weaknesses section.

**Limitations:**

This paper discusses the limitations, but that is not enough.

---

> ### Author Rebuttal · Authors · 2024-08-05
>
> **1a) Improvements regarding Figure 1**
>
> We updated Figure 1 (see Rebuttal Figure 1) to include the latent action prior and highlight the support constraint.
>
> **1b) During the policy training phase, are the imagined trajectories rolled out from the latent state prior  or the latent state posterior ?**
>
> The policy is trained on imagined trajectories generated by the latent state prior. The latent state prior is based on the deterministic transition $f(h_{t-1}, s_{t-1}, a_{t-1})$, which requires actions $a_{t-1}$ in the environment’s action space. Thus, we use the latent action decoder to generate actions from latent actions sampled from the policy.
>
> **2a) Claim that the latent actions generated by the action prior are close to the dataset's action distribution**
> To evaluate the claim we use the following approach:
> For each step in every trajectory within the dataset, we employ k-nearest neighbors to identify the 20 nearest observations and their corresponding actions. We then fit a normal distribution to these actions for each step. Additionally, we sample from the action prior and decoder to generate actions and fit a normal distribution to these for each step as well. Thus, we end up with an approximation of the dataset’s action distribution and an approximation of the action distribution generated by the prior for each step in every trajectory.
> To retrieve a single metric for the whole dataset, we calculate the KL divergence for each step and average over the action dimensions, all steps and trajectories. We conduct the evaluation for two different datasets, resulting in an averaged KL divergence of 0.22 for hopper-expert-v2 and 0.27 for walker2d-medium-replay-v2.
> Moreover, we create an additional figure (Rebuttal Figure 2) to show the aggregated distribution (from leaving the ground to touching the ground) for one exemplary trajectory from the hopper-expert-v2 dataset. The selected trajectory has an averaged KL divergence of 0.33.
>
> **2b) Does the predictive power of using latent actions degrade in performance compared to models using real actions?**
>
> The latent state prior and latent state posterior depend on the deterministic transition $f(h_{t-1}, s_{t-1}, a_{t-1})$. Hence, we use actions from the dataset during model training. This is the same as in a regular RSSM/Dreamer model. Only during policy search, latent actions are decoded and fed into the transition. As a result, the predictive power of our model is comparable.
>
> **3) Differences between the proposed method and Dreamer.**
>
> C-LAP learns a generative model of states and actions $p(o_{1:T},  a_{1:T})$, while Dreamer learns a conditional model $p(o_{1:T} \mid a_{1:T})$. Both learn a policy on imagined trajectories. While Dreamer learns a policy directly in the environment’s action space, C-LAP learns a policy in the latent action space and uses the latent action decoder to generate actions. Dreamer works well for online learning, but fails in the setting of offline reinforcement learning. This can be seen in Figure 6, where “no latent action” corresponds to Dreamer. Dreamer does not restrict the policy to generate actions close to the dataset’s action distribution, hence suffers from value overestimation.
>
> **4) Performance on navigation tasks such as AntMaze?**
>
> We conducted an additional evaluation on D4RL antmaze datasets (Rebuttal Figure 4a). Initially, we did not test our method on these datasets because none of the methods we compare against report results on them. Therefore, we performed a hyperparameter search for MOBILE, varying the penalty coefficient between [0.5, 1.5, 2.5, 3.5] and the rollout steps in [1, 5],  but none of these hyperparameters made MOBILE work on these environments. PLAS is mediocre in the umaze environments but does not work for medium environments at all. For C-LAP we perform a hyperparameter search of $\epsilon$ in [0.5, 1.0, 2.0, 3.0]. Overall, a more extensive hyperparameter search might improve the results for all methods.
>
> In the more challenging diverse datasets and large environments, none of the methods proved effective.These datasets contain numerous suboptimal trajectories, are collected with non-markovian policies and are known to require strong Q-learning, such as Max Q Backup [1] in model-free methods, to be successful. We believe this points to an interesting research direction, as datasets with these characteristics have not been thoroughly explored for model-based approaches.
>
>
> [1] Kumar et al., Conservative Q-Learning for Offline Reinforcement Learning, 2020
>
>
> **5) Why do some experiments in Figure 4, 5 show high returns at the beginning of policy learning (zero gradient step)?**
>
> Please take a look at our general reply.
>
> **6) In table 5 and table 6, the performance of C-LAP is inferior to some baseline models in medium or medium-relay datasets. This results seem to indicate that the effectiveness of the proposed method depends on the quality of the offline dataset.**
>
> Yes, especially the effect of jump-starting policy learning by leveraging the latent action decoder to already achieve high rewards in the beginning of policy training clearly depends on the dataset. We will also add this to the limitations.
>
> **This paper discusses the limitations, but that is not enough.**
>
> To further discuss the limitations of C-LAP, we intend to move lines 214-217 to a separate paragraph in the conclusion. Moreover, we add the following limitations:
> - Depending on the dataset and environment the effectiveness of C-LAP differs.
> - The effect of jump-starting policy learning with the latent action decoder to already achieve high rewards in the beginning of policy training is prominent in narrow datasets, but less effective for diverse datasets.
> - Datasets containing suboptimal trajectories produced by non-Markovian policies, such as those in navigation tasks, pose a challenge for C-LAP
>
> Do you have any further limitations in mind, which we should address?

---

> > ### Comment · Reviewer_6cNB · 2024-08-12
> >
> > Thank you to the authors for their response. Your reply adequately addressed my concerns, and  I will maintain my rating.

---

### Official Review · Reviewer_932j · 2024-07-14

**Soundness:** 2
**Presentation:** 4
**Contribution:** 3
**Rating:** 7
**Confidence:** 4

**Summary:**

The paper approaches the problem of model-based policy learning from static datasets. It firstly identifies that this offline MBRL setup inherits two major concerns from its components: the problem of value overestimation from out-of-distribution actions (common in offline RL) and the bias originated by learning policies from model predictions. In order to address these issues, it proposes C-LAP, a MBRL method that besides learning a latent state space model - as common in the literature - also learns a latent action space from the static dataset. This generative model of states AND actions is further leveraged by the policy optimization objective to constrain the policy actions to share support with the latent action prior, in order to prevent out-of-distribution samples. The paper provides experiments in two offline RL benchmarks, with feature and pixel-based observations, and showing good improvements in the pixel-based observation setting.

**Strengths:**

- The paper addresses the open problem of offline model-based reinforcement learning, which is very relevant to the RL community nowadays. The paper is well-written, very clear and organized: the methodology is carefully and formally described; the baselines are clearly and concisely described (and justified).

- Despite the incremental nature of the paper (while mixing up different building blocks from previous literature), the work is really clever on identifying that a latent modeling of actions would implicitly constrain the action space to the data distribution, which would naturally provide a regularization effect on learning policies with static datasets. Therefore, the general methodology is very well motivated and applied to the problem.

- The asymptotic results in the V-D4RL are really strong and the ablations in Section 4.2 presents very clearly the effect of the proposed method in the problem of value overestimation. These results give solid evidence that the method is indeed addressing the previously raised problems.

**Weaknesses:**

- The claim in L200 regarding a “significant speed up in policy learning” is questionable, since comparing the number of gradient steps to achieve a certain performance is not fair. Some methods can leverage less computation in the updates than others. Also, one can carefully tune the hyperparameters (such as the learning rate, batch size) to minimize the number of gradient steps required, at the cost of more computation. Therefore, showing the gradient steps in the x-axis is not ideal and does not seem to provide useful information. It would be interesting to bring the computational cost for claiming speed up. Optionally, the work could remove such a claim.

- There is an unclear trend in the results of the proposed method: in many cases the starting policy already performs close to the best, while other methods start from performance 0. For instance, refer to the expert cases in D4RL. Why does this happen?
    - The most interesting case is the walker 2d-expert-v2 in Figure 6, where one ablation starts optimal and then goes down. It would be interesting to justify or at least hypothesize what is happening.

- The bound of Eq. 11 is not justified in the paper and looks arbitrary. From my understanding no prior is placed in the policy so this looks like an assumption. It would be interesting to elaborate better about this.

- The last concern is perhaps the most general and crucial to be cleared out during rebuttal. It is clear that the method worked much better for visual domains than feature domains (the results in the D4RL dataset do not show any asymptotic improvement – it is actually worse than MOBILE as per Table 5). In both cases, the action space is the same. It is also low-dimensional. This raises the question: is the model really implementing latent variables for actions or is it actually implementing a hierarchical latent model for the state space? The work explicitly defines “u_t” as latent actions, but it could also be interpreted as a higher-level latent state variable. Given that the gradients from the action decoder backpropagates through both s_t and u_t, I couldn’t find an explanation on why this is not a valid explanation of the method, and this would better justify the discrepancy in the results.

**Minor Concerns/Further Suggestions**

- It would be great to show the value overestimation plots (Figure 6) for the considered baselines, in order to compare the difference among them. It is unclear if they perform worse than the proposed method because of value overestimation or other reason.

- It would also be nice to show a sensitivity analysis of the policy constraint hyperparameter epsilon across a few environments, to help understanding how the proposed method behaves and how much it is environment specific.

- In L105, the paper contrasts the proposed C-LAP to previous works stating that they “rely on uncertainty estimates to generate trajectories within the data distribution”, but I would consider rephrasing that since C-LAP can also be understood as leveraging the uncertainty of the latent action prior to generate trajectories. Perhaps the distinction is that previous methods explicitly use uncertainty estimates, while C-LAP relies on a more implicit constraint.

**Typo:**

- A missing “)” in Eq. 12.

**Questions:**

Please refer to the weaknesses section. Each Major Concern contains questions to be addressed during the rebuttal.

**Limitations:**

- The main limitation in my perspective is that it is not clear why the method presents much stronger results in the visual benchmark (which is supposed to be harder) than in the feature-based benchmark. This opens questions about whether the introduced latent modeling is indeed doing what is hypothesized in the paper.

- There are some limitations in how the results are presented and discussed in the work (please refer to first two bullet points in the Weaknesses section).


**Summary of the Review**:

Overall, the paper is well-written, clearly describes the proposed methodology and offers strong results in the visual benchmarks. The work is based upon a very interesting insight on what latent variables could potentially offer to restrict out-of-distribution actions in offline RL settings. The paper does present some concerns, most related to better explaining the presented results and adjusting claims accordingly. There are also other minor concerns/further suggestions that do not prevent acceptance but would also enrich the paper (which I would appreciate with a higher score). That being said, I believe the paper is already above the acceptance threshold and I am willing to increase my score if my concerns (mostly questions) are properly discussed in the rebuttal phase.







# Post-Rebuttal

Thank you, authors, for properly addressing my questions and concerns during the rebuttal. After rebuttal, I am increasing my score.

---

> ### Author Rebuttal · Authors · 2024-08-05
>
> **The claim in L200 regarding a “significant speed up in policy learning” is questionable, ...**
>
> After discussing pros and cons, we revise the claim and keep gradient steps as we would need to rerun all experiments again to compare computation time (not comparable because of variations in cluster utilizations and nodes) or computational costs in FLOPS (not logged).
>
> Therefore, we revise our claim as follows:
> “...jump-start policy learning by using the action decoder to sample actions that lead to high rewards already after the first gradient steps. This effect is especially prominent in narrow datasets such as expert datasets.”
>
> **There is an unclear trend in the results of the proposed method: in many cases the starting policy already performs close to the best, while other methods start from performance 0. For instance, refer to the expert cases in D4RL. Why does this happen?**
>
> Please take a look at our general reply.
>
> **The most interesting case is the walker 2d-expert-v2 in Figure 6, where one ablation starts optimal and then goes down. It would be interesting to justify or at least hypothesize what is happening.**
>
> In Figure 6, the initial close to best performance is achieved for the same reason as pointed out in the general reply. The ablation “no constraint” degrades in performance thereafter, as the policy is free to generate actions which are not contained in the dataset’s action distribution. Opposing to that, C-LAP restricts the support to the latent action prior with multiples of $\sigma_{\theta}$ centered around $\mu_{\theta}$ (Equation 9), limiting the possibility to sample out-of-distribution actions and thereby restricting value overestimation.
>
> **The bound of Eq. 11 is not justified in the paper and looks arbitrary. From my understanding no prior is placed in the policy so this looks like an assumption. It would be interesting to elaborate better about this.**
>
> We noticed a mistake in text and notation that might have caused confusion. The support of the distribution is chosen to be bounded, not the probability. Thus, we change line 163 to “The support of the policy distribution…” and Equation 11 to:
> $\hat \pi_{\phi}(u_t \mid s_t)$ with  $u_t \in [-1, 1] $
> To explicitly implement the proposed constraint, the support of the policy needs to be bounded between [-1, 1]. Without this bound, using the linear combination in Equation 12 and setting the support with $\hat{\epsilon}$ as multiples of $\sigma_{\theta}$ centered around $\mu_{\theta}$ would not be feasible. Thus, we implement the policy as a TanhNormal distribution.
>
> **Concern: Is the model really implementing latent variables for actions or is it actually implementing a hierarchical latent model for the state space? Why are C-LAP’s results more impressive on images?**
>
> The latent action posterior, prior and decoder can shape the latent state by back-propagating through $z_t$. However, $u_t$ has no effect on state predictions as the latent state prior and latent state posterior are based on the deterministic transition function $f(h_{t-1}, s_{t-1}, a_{t-1})$, which is conditioned on actions $a_{t-1}$. Hence, the only objectives acting on $u_t$ are the action consistency (KL) and action reconstruction loss. Still, some information of $z_t$ could be in $u_t$ as latent action posterior, prior and decoder are conditioned on $z_t$. But to incentivize some kind of hierarchical state in $u_t$, the observation consistency and observation reconstruction loss would need to act on $u_t$, which is not the case.
>
> Coming to the initial concern, there are several possible explanations for why C-LAP’s results are more impressive on images (V-D4RL) compared to full state information (D4RL):
> - The datasets, collected using different policies, and the environments, namely dm-control and gym, differ significantly. For example, the results on halfcheetah-expert-v2 (D4RL) appear less impressive, while the results on cheetah-run-expert (V-D4RL) look great in comparison with the baselines, even though C-LAP achieves 97.1 on halfcheetah-expert-v2 and 36.6 cheetah-run-expert. This makes direct comparisons between D4RL and V-D4RL challenging.
> - The Dreamer architecture commonly performs better on image observations than full state information.
> - The baselines on V-D4RL and D4RL are not the same. With D4RL being the benchmark which attracted more research in the past.
>
> **Value overestimation plots (Figure 6) for the considered baselines**
>
> We created a separate plot to analyze value overestimation for the considered baselines (Rebuttal Figure 3). For PLAS, MOPO, and MOBILE, which estimate Q-values, we calculate the corresponding value estimates by averaging over 10 actions sampled from their respective policies. MOPO and MOBILE have a low value estimate, which can be attributed to the incorporated uncertainty penalties. PLAS’s value estimates are only stable for walker2d-expert-v2 datasets, but collapse for the other considered datasets. Overall, it seems that value overestimation is not the cause for degrading performance for these methods.
>
> **Sensitivity Analysis of the policy constraint hyperparameter epsilon**
>
> We performed a sensitivity analysis (Rebuttal figure 4b) of the policy constraint across all walker2d datasets. Except for the more diverse medium-replay-v2 dataset, adjusting the constraint within the specified range had only a minor impact on the achieved return. Nevertheless, the absence of a constraint leads to a collapse during training (Figure 6).
>
> **Rephrasing L105 “rely on uncertainty estimates to generate trajectories within the data distribution”**
>
> We will rephrase it as follows to highlight the distinction:
> “Unlike other model-based offline reinforcement learning methods that learn a conditional model $p(o_{1:T} |a_{1:T})$ and rely on ensemble based uncertainty penalties on the Bellman update to generate trajectories within the data distribution, …”

---

> > ### Comment · Reviewer_932j · 2024-08-12
> >
> > Dear authors,
> >
> > Thank you for the detailed explanations and additional empirical evidence. My concerns were addressed, and I am increasing my score accordingly.

---

### Official Review · Reviewer_UrFS · 2024-07-24

**Soundness:** 3
**Presentation:** 4
**Contribution:** 3
**Rating:** 6
**Confidence:** 4

**Summary:**

The work introduce C-LAP (Constrained Latent Action Policies), a novel approach to model-based offline reinforcement learning in POMDPs. Notably, C-LAP does not employ explicit reward penalty terms regarding the action space.

The paper proposes a methodology for learning latent variables for both states and actions based on the state-space models (SSMs). It then frames policy optimization as a constrained optimization problem, aiming to maximize N-step returns while ensuring that latent actions remain within the support of the latent action prior.

This approach allows the policy to maximize returns predicted by the model without incorporating reward penalty terms, such as uncertainty penalization. Empirically, C-LAP demonstrates superior performance compared to baseline methods that incorporate uncertainty penalties in their reward structure.

**Strengths:**

The manuscript presents a novel and effective model-based method for offline reinforcement learning in POMDPs. Prior approaches in the domain of model-based offline RL often incorporate penalty terms to mitigate OOD actions and prevent overestimation of the value function. This penalization, however, can restrict the maximization of value functions related to possibly promising actions.

In this work, the authors extend a standard state-space model (SSM), exemplified by model-based approaches such as Dreamer, to include latent action variables. By constraining the latent action space to adhere to an action prior, the generation of OOD actions is effectively precluded.
This modification is both simple and effective, addressing a critical challenge in the field.

The effectiveness of the proposed method is validated across both MDP and POMDP settings using standard benchmarks such as D4RL and V-D4RL. Moreover, the authors provide a thorough ablation study to identify the critical components contributing to performance improvements. As described in Figure 6, omitting the constraint on the latent action prior is the most crucial to performance, underscoring its significance as a key contribution of this research.

**Weaknesses:**

Despite the novelty of this work, the motivation for adopting latent action spaces is insufficiently articulated.
The paper does not clearly address why constraining with respect to the latent action space is advantageous over the actual action space.
Specifically, it remains unclear whether C-LAP offers any benefits compared to SSMs that include a support constraint term directly on the actual action space, as expressed by the constraint  $\mathbb{E}_{s \sim p, a\sim \mu(s)}[\pi (a|s) ] \ge \epsilon$
(where $\mu$ denotes the behavior policy) for an instance.
I believe that a more thorough explanation for the motivation could significantly strengthen the contributions and arguments of this work.

(Minor comment) In Eq. (7), a constraint term would be a typo. It would be more accurate to revise this as $\mathbb{E}_{p,\pi}[p(h_t|s_t)]\ge \epsilon$, given that $h_t$ is sampled from $\pi(s_t)$.

**Questions:**

Q1. The manuscript suggests potential applicability of the C-LAP framework beyond the traditional models explored. Have the authors considered leveraging foundation models, such as transformers or diffusion models, within the latent space formulation of C-LAP? If so, could you elaborate on how C-LAP might integrate with these advanced model architectures?

Q2. The proposed C-LAP method appears to be promising for environments characterized by high-dimensional action spaces, including those analogous to language action spaces. Could the authors provide insights into the adaptability of C-LAP to such high-dimensional scenarios? What modifications, if any, would be necessary to optimize its performance in these contexts?

**Limitations:**

The authors included their limitation in Section 4.2. They address societal impacts of their work in the checklist #10. Broader Impacts.

---

> ### Author Rebuttal · Authors · 2024-08-05
>
> **The paper does not clearly address why constraining with respect to the latent action space is advantageous over the actual action space. Specifically, it remains unclear whether C-LAP offers any benefits compared to SSMs that include a support constraint term directly on the actual action space, as expressed by the constraint  (where  denotes the behavior policy) for an instance. I believe that a more thorough explanation for the motivation could significantly strengthen the contributions and arguments of this work.**
>
> To implement a support constraint alongside a state-space model, it is necessary to estimate the behavior density. This can be achieved by training an additional VAE, as demonstrated in SPOT [1]. Conversely, C-LAP integrates the objectives of state-space modeling and behavior density estimation into a single generative model of observations and actions.
> Using a regular state-space model combined with a behavior density estimator, one could implement a support constraint directly on the actual action space. However, this approach presents the following challenges:
> - If the policy generates actions directly in the environment’s action space, the generative action decoder cannot be used to jump-start policy learning. This drawback becomes especially clear in the "expert" datasets shown in Figures 4 and 5, as discussed further in the general rebuttal section. Another approach is to learn a latent action policy and use the generative action decoder, similar to our C-LAP and PLAS [2], while implementing the constraint on the actual action space.
> - However, both of these variants require the introduction of a Lagrange multiplier in the policy objective. A Lagrange multiplier does not guarantee compliance with the constraint and is difficult to tune and analyze. C-LAP avoids altering the policy objective and uses an explicit parameterization of the latent action space through the latent action prior. The parametrization has a clear interpretation (multiples of $\sigma_{\theta}$ centered around $\mu_{\theta}$) and is not very sensitive to the choice of $\epsilon$ (Rebuttal Figure 4b).
> - Implementing a support constraint via a Lagrange multiplier $-\lambda \log \pi_{\beta}$ not only restricts the support, but also influences the whole shape of the distribution. Explicit parameterization on the other hand only scales the distribution to ensure the boundaries.
> - The behavior density on the actual action space needs to be approximated with multiple samples from the latent action prior and subsequent decoding, while a support constraint on the latent action space does not require sampling from the latent action prior at all.
>
> **Q1. The manuscript suggests potential applicability of the C-LAP framework beyond the traditional models explored. Have the authors considered leveraging foundation models, such as transformers or diffusion models, within the latent space formulation of C-LAP? If so, could you elaborate on how C-LAP might integrate with these advanced model architectures?**
>
> Yes, as some foundation models are optimizing a similar objective as $p(o_{1:T}, a_{1:T})$, leveraging these models is an interesting direction for future research. C-LAP explicitly defines both a latent state space and a latent action space through the latent action state-space model formulation. This formulation is applicable to other model architectures. Therefore, the encoder-prior-decoder structure of states and actions in C-LAP can be replaced with a transformer or a diffusion model to learn a foundation model with an explicit internal structure. Moreover, given an already pretrained foundation model, a couple of interesting research directions come to our mind:
> - C-LAP can be used to adapt the foundation model’s action space. For example, given a generalist robot policy such as Octo [3], it is feasible to use C-LAP to learn a new action head while ensuring state space consistency.
> - C-LAP can be used to distill knowledge from a foundation model into a smaller model by using the foundation model as a differentiable environment.
> - Foundation models optimizing $p(o_{1:T} \mid a_{1:T})$ can replace the latent state prior, latent state posterior and observation decoder. By fine-tuning with C-LAP on a small dataset, we can use the predictive power of the foundation model and only learn the additional latent action representation.
>
>
> **Q2. The proposed C-LAP method appears to be promising for environments characterized by high-dimensional action spaces, including those analogous to language action spaces. Could the authors provide insights into the adaptability of C-LAP to such high-dimensional scenarios? What modifications, if any, would be necessary to optimize its performance in these contexts?**
>
> When adapting C-LAP to high-dimensional action spaces, two aspects should be considered: First, the size of the latent action space must be adjusted to suit the environment. Second, it can be beneficial to condition the deterministic transition $f(h_{t-1}, s_{t-1}, a_{t-1})$ directly on latent actions $u_t$​ instead of decoded actions $a_t$​. This approach simplifies the learned latent state prior $p_{\theta}(s_{t} \mid s_{t-1}, u_{t-1})$ by reducing the dimensionality of the considered actions.
> To adjust C-LAP for action spaces informed by language, we propose to extend the generative model to $p(o_{1:T}, a_{1:T} | L_{1:T})$ with $L_{1:T}$ as a language condition. Deriving a state-space formulation, the latent action posterior, latent action prior and action decoder could be extended with an additional language condition similar to “LILA: Language-Informed Latent Actions” [4]
>
> ---
>
> [1] Wu et al., Support Policy Optimization for Offline Reinforcement Learning, NeurIPS 2022
>
> [2] Zhou et al., PLAS: Latent Action Space for Offline Reinforcement Learning, CoRL 2020
>
> [3] Ghosh et al., Octo: An Open-Source Generalist Robot Policy, 2024
>
> [4] Karamcheti et al., LILA: Language-Informed Latent Actions, CoRL 2021

---

> > ### Comment · Reviewer_UrFS · 2024-08-11
> >
> > Thank you for your detailed responses to my concerns. I will maintain my rating and encourage the authors to incorporate the discussed details on the motivations into the revision.

---

### Author Rebuttal · Authors · 2024-08-05

First and foremost, we want to thank all reviewers for their time, effort, and insightful feedback! This clearly helped us to improve our paper!

**We conducted further experiments and added the following figures:**

- Updated version of Figure 1 to include the latent action prior and highlight that the policy is generating actions within the support of the latent action space. (Rebuttal Figure 1)
- Comparison of the dataset's action distribution to the distribution of actions sampled from the latent action prior and decoder. (Rebuttal Figure 2)
- Analysis of value-overestimation for all baselines (Rebuttal Figure 3)
- Evaluation of AntMaze environments (Rebuttal Figure 4a)
- Sensitivity analysis of the support constraint parameter epsilon on the walker2d datasets (Rebuttal Figure 4b)

Moreover, we noticed multiple reviewers wondering why C-LAP’s performance is sometimes close to the highest reward already in the beginning of policy training (Figure 4 and Figure 5).

**Why is C-LAP’s performance sometimes close to the highest reward already in the beginning of policy training?**

The model is trained to generate actions contained in the dataset’s action distribution. If the dataset is narrow, generated actions when sampling from the latent action prior will fall into the same narrow distribution. For instance, in an expert dataset, sampled actions will also be expert-level actions. During policy training, instead of sampling  from this prior, we  restrict the support of the policy dependent on the latent action prior. Thus, sampled latent actions from the policy will always be decoded to fall into the dataset’s action distribution. So even a randomly initialized policy in the beginning of the training can generate a high reward by using the latent action decoder. This effect clearly diminishes if the dataset’s action distribution is less narrow, which can be seen in the extremes, e.g., when comparing medium-replay to expert datasets in Figure 4. Thus, policy training is more essential for medium-replay datasets to only sample actions leading to high rewards.

---

### Comment · Area_Chair_ynSm · 2024-08-11
**Please respond to author response if you haven't**

Dear Reviewers,

Before the author-reviewer discussion period ends, please make sure that you have read the author responses, acknowledge that your review reflects them, and engage with the authors if necessary.

Thank you!

---

### Decision · Program_Chairs · 2024-09-25

**Decision:**

Accept (poster)

**Comment:**

This paper introduces Constrained Latent Action Policies (C-LAP), an approach for model-based offline RL designed to handle value overestimation. C-LAP employs a latent action state-space model to generate actions that are constrained within the support of a learned action distribution, mitigating the risk of generating out-of-distribution actions. By formulating the policy learning as a constrained optimization problem, C-LAP eliminates the need for additional uncertainty penalties, thereby speeding up the learning. The approach is tested on benchmarks with both low-dimensional and visual observations, demonstrating competitive performance, particularly in visual tasks. In summary, C-LAP demonstrates a robust solution to the challenges of offline reinforcement learning by maintaining close distances to the original data distribution while enabling efficient policy training.